# Synthesis, Characterization, and Stability Assessment for the Benzoate, Hydrochloride, Malonate, and Nicotinate Salts of Bedaquiline

**DOI:** 10.3390/ph16020257

**Published:** 2023-02-08

**Authors:** Mercy A. Okezue, Stephen R. Byrn

**Affiliations:** Industrial and Physical Pharmacy Department, Purdue University, West Lafayette, IN 47907, USA

**Keywords:** lead candidate, solubility, X-ray diffraction, Rietveld refinement, bioavailability, NMEs, DSC, TGA, water sorption, single crystal

## Abstract

Bedaquiline has been approved as a combination therapy to treat multi-drug-resistant tuberculosis in adults ≥ 18 years old. The citrate, fumarate, phosphate, and tartrate salts have obtained patents, but the structures for these moieties have not been extensively described in the literature; only the powder X-ray patterns have been published. To expand the knowledge of the bedaquiline structure, this study provides detailed information for the synthesis, elucidation, characterization, and stability of four additional new potential molecular entities, namely, benzoate, hydrochloride (HCl), nicotinate, and malonate salts. The salts were formed using a 1:1 ratio of the counter ions (acids) to a 30 mg equivalent of the bedaquiline free base. The principles of the International Conference on Harmonization Q6 were used to characterize the new salts and their stability-indicating parameters were evaluated at 0, 3, and 6 months under accelerated conditions of 40 °C and 75% relative humidity. The benzoate salt exhibited the lowest tendency to lose its chemical potency. Aside from the HCl salt, the others retained their chemical structure, displaying long-term stability. All salts were non-hygroscopic and the hydrated benzoate and nicotinate salts were stable to dehydration. Regarding their chemical potencies, thermal analysis, chemical stability, and water sorption potential, the salts were ranked as follows: benzoate > malonate > nicotinate > HCl.

## 1. Introduction

Bedaquiline is one of the two most important drugs for treating tuberculosis. It is used as a combination therapy to treat multi-drug-resistant tuberculosis (TB) in adults ≥ 18 years old and was approved by the US FDA on 28 December, 2012. The fumarate salt, marketed as Sirturo^®^, is the commercially available option [1]. According to the information available online, the US patents for the two fumarate formulations expire in 2026 and 2029. However, the patent WO 2004/011436 for the free base will expire on 18 July, 2023 [2].

Previous attempts have been made to develop salts of bedaquiline. The citrate, phosphate, and tartrate salts have obtained patents, but the structures for these moieties were not extensively described in the literature; only the powder X-ray patterns have been published. However, our research laboratory generated single crystals, which were solved, for the fumarate, benzoate, and maleate salts [3,4]. The published data fostered a better understanding of the physicochemical properties of bedaquiline, which is critical for developing formulations for TB treatment in animal and clinical studies. This paper highlights the synthesis and characterization of additional salts with improved stability and other potential advantages over the fumarate salt. In principle, the new salts can be approved as new molecular entities (NMEs).

As is the case with new drug substances, it is important to determine the physicochemical properties of NMEs. The International Committee on Harmonization (ICH) Q6 guidance describes the general and specific tests that should be conducted to provide primary specifications to control the quality of new substances and NMEs. They form part of the control strategy to ensure consistency in manufacturing an API or NME [5]. The guidance recommends general tests such as assays, identification steps, and the determination of impurities. In addition, other specific tests are also required in order to understand the thermal, solubility, water-sorption, and polymorphic properties, depending on the nature of the new compound.

Thermal analyses are concerned with the temperature-related physical and chemical properties of a compound. Instrumental methods provide information on solvation, dehydration, decomposition, melting points, purity, crystallinity, and the transitions between polymorphic forms of compounds. A common instrument used for measuring temperature transitions is the differential scanning calorimeter (DSC), which provides information on melting points. The changes in the mass of a material in response to increasing temperatures are often assessed with the use of thermogravimetric analysis (TGA) equipment. DSC and TGA have been widely used as thermal analysis techniques in characterizing crystalline pharmaceutical materials [6]. The physical transformations, e.g., melting, polymorph conversion, etc., that occur when compounds are heated in a DSC are accompanied by changes in their enthalpies. These changes are recorded either as endothermic or exothermic events in the DSC thermographs. The changes in mass that occur during a TGA provide information on events related to dehydration/desolvation, as well as material decomposition. Accurate measurements of these parameters describe characteristics of pharmaceutical compounds and can be used in their characterization [7]. In this study, we determined the thermal properties of bedaquiline salts.

Hygroscopicity is used to assess the amount of moisture that can ingress into a pharmaceutical compound at a fixed temperature, depending on the relative humidity of the environment. The presence of moisture can affect the activity of a compound, altering its physical and chemical properties [8]. To access the effect of moisture, a common test of the material’s propensity to pick up water—its hygroscopicity—is generally recommended as one of the criteria for selecting a lead candidate. The results of these tests guide the selection of excipients during product formulation; manufacturing, packaging, and storage conditions; and finally, assessments of the need for water corrections in analytical assays [9]. In this study, we used different humidity chambers containing saturated solutions of salts described in USP/NF general chapter <922> on water activity to measure the sorption tendencies for the four new bedaquiline salts.

Solubility plays an important role in determining the best formulation for a drug molecule, the selection excipients, and in predicting the drug’s bioavailability. The aqueous, ideal, and non-ideal solubilities of the new benzoate salts were extensively described in one of our earlier publications [10]. In the study presented here, we focused on generating data for bedaquiline salts that are critical to their quality attributes, namely, their identification, assay results, melting points, particle sizes, polymorphic forms, and water content.

The research question for this study was ‘how can new salts of bedaquiline be developed as alternatives to the fumarate salt?*’* and our hypothesis was that mixing equimolar amounts of a bedaquiline base with select counterions that had ≥2 pKa units, in organic solvents, would yield salts. To address this query, two objectives were developed:

**Objective 1:** 
*Provide methods for synthesis and elucidate the structures for the new salts of bedaquiline.*


**Objective 2:** 
*Generate data to create preliminary specifications for the new bedaquiline salts.*


## 2. Results

### 2.1. Methods for Synthesis of Five New Salts of Bedaquiline

The solvent solubility results showed that acetone (<1 mL), acetonitrile (~5 mL), ethanol (~6 mL), ethyl acetate (~1.5 mL), methanol (~8 mL), 2-propanol (~9 mL), and tetrahydrofuran (<1 mL) dissolved 30 mg of bedaquiline. Five new salts were synthesized by dissolving 1:1 ratios of the individual salt formers and the bedaquiline base (30 mg, 0.054 millimoles, pKa (basic) 8.91) in three solvents. The equivalent amounts of acids used in the various experiments are shown in Table 1. The salicylate, malonate, and nicotinate salts were synthesized using slow evaporation after dissolving the acids and bedaquiline in a 1:1 mixture of acetone and 2-propanol, whereas acetone was used to precipitate the hydrochloride (HCl) salt. However, the HCl salt disproportionated to the free base in 2-propanol. The pKa values for the salt formers were obtained from the PubChem database, whereas bedaquiline pKa is available in DRUGBANK online. Crystalline salts of bedaquiline were formed from the 1:1 stoichiometric acid-base mixtures.

### 2.2. Determination of the Chemical Structures for the Salts and Confirming Identity via Spectrometric Techniques (Achieved Study Objective 1)

#### 2.2.1. Single Crystals and Powder X-ray Diffractometry Determinations

Single-crystal X-ray determinations were used to elucidate the chemical structures of the salts. Further structural confirmations were achieved using Rietveld refinement of the salts’ powder X-ray diffraction (PXRD), matched with the corresponding single crystals. An important strategy in this study involved the use of optical microscopy to hand-isolate crystals that formed from the various evaporations. Even when a complex mixture of salts and solvates was formed by crystallization, single-crystal determination provided a direct confirmation of one of the species present; see Figure 1. The intensity of the diffraction patterns as a function of characteristic 2θ diffraction angles is shown in Figure 2. The characteristic 2θ peaks of the new salts generated distinctive fingerprints or patterns which could be used to identify them. The experimental details for elucidating the structures of the single crystals for bedaquiline nicotinate and malonate are found in Appendix A
Table A2 and Table A3; others were published in our earlier communication (*Crystal structures of salts of bedaquiline*). The new structures were deposited with the CCDC and allocated the following numbers: nicotinate (Deposition #2179234), hydrochloride (2179235), benzoate (2036004), and malonate (2179236) [10].

The salicylate salt yielded a PXRD that indicated that salt formation was successful. However, all experiments that were set up for growing the single crystals yielded blobs of white masses that could not be elucidated.

#### 2.2.2. Rietveld Refinements

The single crystals were matched against each salt’s PXRD to confirm if the salts were successfully formed from the stoichiometric reactions between the bedaquiline free base and the acids (salt formers). The characteristic 2θ peaks of the new salts matched the patterns generated from their single crystals. Rietveld refinements were used to demonstrate the level of overlap of the salts’ fingerprints on their single crystals; see Figure 3. Other chemical properties of the new salts of bedaquiline are summarized in the Appendix A, in Table A4.

#### 2.2.3. Raman Spectrometry

This technique is used to assess changes in polarizability based on a salt’s ability to provide information for the measurement of Raman scattering. The scattering is directly related to the frequency of molecular vibrations which give rise to bands that can be used to detect functional groups, and the fingerprint region that is used to identify compounds [6]. However, the Raman spectra generated from the bedaquiline salts were not sufficiently strong to generate bands that could enable us to distinguish between the salts and the free basel see Figure 4. Therefore, Raman spectroscopy was not a good spectral technique to identify the bedaquiline salts.

### 2.3. Generated Data to Create Preliminary Specifications for the New Bedaquiline Salts (Achieved Study Objective 2)

#### 2.3.1. Assay/Purity and Solubility Determinations by HPLC

The HPLC chromatograms for the new salts gave similar retention times to those of the reference standard (bedaquiline base) and a control sample (fumarate salt); see Figure 5. The samples were analyzed with the conditions listed in the Materials and Methods section.

The results of the assay for assessing bedaquiline content were calculated using Equation (1) in the Materials and Methods section, comparing the peak areas for similar concentrations of the bedaquiline standard (free base) and the new salts. Table 2 summarizes the purity determinations for approximately 50–100 µg/mL concentration solutions of the salts in methanol, using a method validated for the HPLC analysis of bedaquiline [11].

Solubility experiments were carried out in water and simulated gastric (0.01 NHCl) and intestinal fluid (pH 6.8 phosphate buffer). The aqueous solubilities were determined at 24 to 72 h intervals and the results could be ranked as follows: hydrochloride (0.6437 mg/mL) > malonate (0.0268 mg/mL) > nicotinate (0.0024 mg/mL) > benzoate (0.0004 mg/mL). These showed improved aqueous solubility over the free base. Extensive details of the results were published in our communication for the solubilities of the four new salts of bedaquiline [10].

#### 2.3.2. Particle Size and Shape Analysis

Particle size affects the rate of dissolution, and therefore the safety and performance of a drug product. ICH Q6 emphasizes the importance of controlling particle sizes, especially for poorly soluble drugs such as bedaquiline. Particle shape accounts for some subtle differences that can occur between samples with respect to the translation to a circle-equivalent or spherical-equivalent diameter. Differences in particle shape can impact product parameters such as flowability, abrasive efficiency, and bioavailability [12]. The Malvern Morphologi G3 instrument used for determining particle size for the bedaquiline salts uses high-magnification microscopy as a technique to analyze particle images. Table 3 summarizes the parameters measured in order to determine the bedaquiline salts’ particle size distribution, and details of the circular equivalent diameter measurements for the salts are provided in Figure A1 in the Appendix A.

The particle size distributions were determined based on the particle size values D_10_, D_50_, and D_90_ and were calculated by means of the span = [D_90_ − D_10_]/D_50_ [13]. The span is an indication of how far apart the 10 percent and 90 percent points are, normalized with the midpoint. The narrower the distribution, the smaller the span value. The particle size distribution for the benzoate and hydrochloride salt was narrower than those of the other salts. The tendency for powders to segregate is lower when span values are smaller.

A confocal microscope was used to elucidate the morphology of the salts. Figure 6 shows the shapes of the bedaquiline salts. Although measurements from the confocal microscope were not used to determine the particle sizes, the 10 µm scale observed for the images generated for the HCl and salicylate salts were in agreement with the measurements obtained for the Morphologi G3 maximum circle equivalent diameters for the salts. However, the researchers acknowledge that sampling errors may also have affected the span values obtained using the Morphologi G3 instrument.

A smaller standard deviation between the particle sizes would translate to a narrower difference in the particle size distribution for the bedaquiline salts, as well as a lower span value.

#### 2.3.3. Thermal Analysis

The thermal analysis of bedaquiline salts provided information about their melting points and the presence of volatile solvents and hydration states. Salts that exhibit low melting points are candidates for exclusion from further drug development.

##### Melting Point/Range Data

The melting points of the bedaquiline crystalline salts were determined using a Thomas Hoover capillary melting point apparatus; the uncorrected values are documented in Table 4 below. The malonate and hydrochloride salts had higher melting points than those of the nicotinate and benzoate salts. However, the free base had the highest melting point.

##### Differential Scanning Calorimetry (DSC)

DSC was used to measure the differential heat flow in the salts, as the samples were scanned at various temperatures. Indium was used as the reference sample for calibrating the equipment prior to sample analysis. In this heat-flux DSC, the heat differential between the bedaquiline salt samples and the reference pin-holed pan was determined. During endothermic events, the heat absorbed by the sample is higher than that of the reference, whereas exothermic transitions are observed to occur when the heat flow to the sample is lower than that of the reference. The thermograms in Figure 7 were generated in the DSC analyses of the benzoate, malonate, nicotinate, and hydrochloride salts, which were heated from 25 °C to 250 °C at the rate of 10 °C/min. The endotherms at ~109 °C, 124 °C, 155 °C, and 172 °C reflect the melting transition temperatures for the nicotinate, benzoate, malonate, and hydrochloride salts, respectively. The DSC results for the salts were close to the data obtained from the determination of the melting point using the Thomas Hoover capillary apparatus.

##### Thermogravimetric Analysis (TGA) of the Hydrated Benzoate and Nicotinate Salts

TGA was conducted to measure the changes in the mass of the bedaquiline (BQ) salts as a function of time over 25 °C to 200 °C at a heating rate of 10 °C/min. A Perkin Elmer Pyris TGA 400 Analyzer was used to generate the data for the bedaquiline salts shown in Figure 8. The experiment monitored the loss in mass that occurred as a result of desolvation, where adsorbed or bound solvents left the salts at elevated temperatures, resulting in a decrease in the mass of the benzoate and nicotinate salts. Bedaquiline nicotinate lost ~0.25 mg of its weight when the sample was heated up to 140 °C. Similarly, bedaquiline benzoate lost ~0.30 mg of its weight when the sample was heated up to 140 °C. The losses in mass for the two salts were comparable to the moisture content results obtained in the Karl Fischer analysis.

#### 2.3.4. Hygroscopicity

The long-term hygroscopicity was monitored by exposing selected samples to both 75% relative humidity (RH) and 0% RH conditions for up to 5 months. Small chambers were set up, containing either a saturated sodium chloride solution (75% RH) or Drierite (0%RH). The short-term exposure lasted for 3 weeks.

The amount of weight gain or loss was calculated based on the initial sample weight prior to exposure to these conditions [14]. The malonate and hydrochloride salts gained more than 2% of moisture, whereas the other bedaquiline salts investigated were non-hygroscopic (<2% weight gain) upon exposure to 75% RH and 0% RH under ambient conditions for 150 days; see Figure 9.

The short-term exposure to conditions of increasing RH suggested that at 25 °C, no salt gained more than 0.02% *w*/*w* at 92.3% RH. The hydrochloride and malonate salts did not exhibit similar weight gains in the long-term sorption/desorption experiments. The benzoate and nicotinate results were similar to the values observed for long-term exposure (Figure 10).

#### 2.3.5. Water Content, Determined via Karl Fischer Titration

Karl Fischer titration was used to estimate the amounts of residual water, expressed as % *w*/*w*, in the hydrated salts. The weights of the samples titrated were about 50 mg and 30 mg of the benzoate salts from the 2-propanol-water and acetone solvents, respectively. The benzoate salt formed from 2-propanol with water as the antisolvent contained 3.37% moisture, whereas the 1.17 hydrated form contained 3.33% water. The nicotinate salt contained 3.53% water.

#### 2.3.6. Polymorph Salt Screening Experiments

None of the results of the polymorph salt screening for the benzoate salt revealed a new crystal form. This suggests that no new polymorph was formed at the 10 kg laboratory scale for the benzoate salt. The full details of the polymorph screening results for bedaquiline benzoate were published in an earlier communication [15].

### 2.4. Stability

Stability testing was carried out on the new molecular entities to determine how their quality characteristics were affected by environmental factors such as humidity and temperature. Stability testing generally provides data that are used to establish retest periods for active pharmaceutical products (APIs) and expiry dates for finished pharmaceutical products, as well as their storage conditions. For a 6 month accelerated stability study for APIs, a minimum of three points, including the initial and final time points (e.g., 0, 3, and 6 months), is recommended in ICH Q1A (R2). In this study, in the protocol for stability studies we assessed the physical appearance, PXRD, DSC, TGA, and potency of the new bedaquiline salts.

#### 2.4.1. Physical Appearance of Stability Samples

The nicotinate salt was off-white, whereas all the other salts were white crystalline powders at time 0. At the end of 3 and 6 months, none of the stability samples for any of the bedaquiline salts showed any visible change in physical appearance.

#### 2.4.2. PXRD of Stability Samples

The chemical structures of benzoate, nicotinate, and malonate were not significantly altered at 3 or 6 months. The PXRD values of the salts were refined against their single crystals when examined with the Rietveld refinement technique. However, the hydrochloride salt lost its crystallinity at 3 months, thereby altering its chemical structure as observed in the PXRD fingerprints. The diffractograms for the salts are shown in Figure 11.

#### 2.4.3. Potency at 0, 3, and 6 Months Accelerated Stability

The chemical potencies of the accelerated samples were calculated and are summarized in Figure 12. The potency of benzoate declined from 90.4% at the initial time point to 86.1% at the 6th month. That of malonate declined from 95.6% to 71.3%, that of nicotinate from 84.7% to 64.1%, and that of hydrochloride from 101.9% to 77.3%.

#### 2.4.4. Thermal Analysis of Stability Samples

The TGA analysis of the hydrated benzoate and nicotinate salts after 6 months of accelerated stability conditions suggested that the salts did not become anhydrous. The positions of the thermograms at 6 months were similar to those observed at the 0-month starting point; see Figure 13.

Thermographs of the salts’ DSC 6-month stability samples did not show any significant changes in the positions of the endotherm patterns for benzoate or malonate. However, the 6-month thermograph for nicotinate did not show the multiple vortexes that were observed for the initial sample at 0 months; see Figure 14.

## 3. Discussion

### 3.1. Salts of Bedaquiline

pH adjustments were used to form salts in situ when equimolar amounts of counterions (acids) reacted with the free base of bedaquiline. The flexible backbone of the bedaquiline moiety made it amenable to the formation of salts with counterions that satisfied the pKa rule. The salts were formed from a 1:1 stoichiometric ratio of the acids and the bedaquiline base. All the acids that successfully formed salts had a pKa of more than 2 pKa units less than that of bedaquiline, 8.61. This research successfully elucidated the synthetic procedures for the benzoate, malonate, nicotinate, salicylate, and hydrochloride salts. However, none of the efforts to form single crystals of the salicylate salts were positive, even when the experiments were left undisturbed for more than 6 months in different solvent and vapor diffusion systems. Therefore, the salicylate salt was not characterized further. On the other hand, the benzoate salt had the highest probability for success because it was possible to synthesize it using several routes, which have been extensively discussed in two earlier publications [3,15] The malonate and nicotinate salts were successfully synthesized via the slow evaporation of a 1:1 mixture of acetone and 2-propanol.

The hydrochloride salt was prone to disproportionation when 2-propanol was used as the solvent. This phenomenon may be explained by two plausible reasons. The assumption was that 2-propanol raised the pH of the reaction milieu such that it was greater than the pHmax, thereby making the bedaquiline free base the species at saturation equilibrium. Another assumption was that the use of acetone reduced the pH, causing it to be lower than the pHmax so that the hydrochloride salt was the solid species at equilibrium. These assumptions were based on earlier experiments that described the principles of salt formation and the interrelationships of pH a solubility [16]. Alternatively, the disproportionation of the hydrochloride salt may be related to its higher aqueous solubility (0.6437 mg/mL) [10], which would lower the pHmax value and make it more prone to conversion to the free base with increased environmental pH [17]. If the goal was to pursue the hydrochloride salt, it would have been necessary to conduct further studies to fully understand the physical and chemical properties of the salt, relating its pHmax and solubility to its tendency to disproportionate. A pharma R&D would take care in controlling the pH of the reaction milieu to avoid the interconversion of the hydrochloride salt to the bedaquiline free base. This is especially important, as the FDA requires that manufacturers use the guidance of Pharmaceutical Development (ICH Q10) to monitor the control space for their formulation and ensure that only the desired (approved) drug form is existing in the marketed product.

#### 3.1.1. Solubility of the New Bedaquiline Salts

Forming salts of a poorly aqueous molecule may improve its oral [16]. The new salts of bedaquiline showed higher solubilities in different biorelevant media compared to that of the free base. A low value of 0.0000193 mg/mL was reported in the DrugBank database for the bedaquiline molecule. However, the malonate (0.0268 mg/mL), nicotinate (0.0024 mg/mL), HCl (0.6437 mg/mL), and benzoate (0.0004 mg/mL) salts showed higher aqueous solubilities [10].

#### 3.1.2. Particle Size Distribution and Particle Shapes for the Salts

Particle shape is important in understanding the flow properties of powders in additive pharmaceutical manufacturing. The benzoate and hydrochloride salts exhibited the lowest span values, at 3.7 and 2.5, respectively, an indication that they will exhibit lower tendencies towards segregation during powder flow in a compression hopper. This knowledge is important when selecting the particle size of other excipients in the salts’ formulations. Previously, the contributions made by particle shape were not extensively elucidated in pharmaceutical manufacturing. However, with the introduction of additive manufacturing in 3-D technology, there is a need to have a better understanding of particle shape as materials are layered on top of each other to form a product. Some studies have evaluated how shape affected the packing density and roughness of powder beds and suggested that when lower recoating velocities were used, elongated particles exhibited better flow characteristics. In contrast, at higher velocities, spherical particles were more desirable [18,19]. Furthermore, heat transfer within a powder bed was highly correlated with particle shape [20]. These studies underscore the importance of particle shape in characterizing NMEs.

#### 3.1.3. Thermal Analysis of Bedaquiline Salts

Transitions from one state of matter to another occur at defined conditions of temperatures. Since these transitions occur at characteristic temperatures, the accurate determination of melting and boiling points can be used to create specifications to identify a substance. In this study, there were some correlations between the melting points obtained using the capillary apparatus and the endotherms observed in the DSC analysis of the bedaquiline salts. The melting ranges obtained using the Thomas Hoover capillary apparatus for malonate (154 °C ± 1 °C) were close to the DSC 155 °C melt results. The hydrochloride salt (163 °C ± 1 °C) exhibited an endotherm at 173 °C. However, the benzoate salt, depending on the solvent used for synthesis, exhibited a capillary melt value between 128 °C and 134 °C, but the DSC endotherm suggested that after the water loss event, the salt melted at about 116 °C. Similarly, the nicotinate salt with a capillary melt of about 131 °C had an unexpected endotherm at 101 °C and other DSC dips near 130 °C. For the nicotinate salt, the initial endotherm likely occurred due to the loss of its water of hydration from the crystal lattice. Based on the capillary method melting range, a somewhat ill-defined endotherm around 130 °C seemed to represent the DSC melting of the nicotinate salt. The presence of two major endotherms in the benzoate, hydrochloride, and nicotinate salts confirmed that they had either hydrates or solvate components represented in the first endotherm events in each DSC thermogram. The presence of these two major endotherms suggested that the salts retained their hydrated forms after 6 months of accelerated stability conditions at 40 °C and 75%RH. Furthermore, although there were also no observable shifts in the DSC endotherms for the salts’ melting points, some additional endothermic events were seen. These peaks were probably attributable to the presence of impurities in the bedaquiline salts, which were not synthesized under GMP conditions.

The presence of impurities has been shown to affect the melting points of compounds, analogous to the results reported for differences in DSC endotherms from estradiol implants with different levels of impurities [21]. The USP/NF general chapters on thermal analysis <891> confirmed that differences observed between the salts’ capillary melting points and the DSC melts were expected. The compendium documents that it is somewhat difficult to objectively reproduce the “onset” and “peak” temperature in a DSC endotherm; the difference between the two is related to the purity of the compound. Therefore, there may not be a good correlation between the melting points obtained using the two methods. Other limitations of thermal analysis are associated with solid solution formation, the insolubility of salts in the melt, te possibility of polymorphism, and decomposition during the heating phase of analysis [7].

The thermographic analysis (TGA) of the salts was carried out to assess the weighed masses as a function of increasing temperature and to investigate the possible dehydration/desolvation processes, as well as the decomposition of the salts. The benzoate (C_32_H_32_BrN_2_O_2_, C_7_H_5_O_2_, 1.166(H_2_O)) and nicotinate (C_32_H_32_ BrN_2_O_2_, C_6_H_4_NO_2_, C_3_H_8_O, H_2_O) salts lost approximately 0.30 mg and 0.25 mg of their weights, respectively, as the samples were heated up to 140 °C. These weight losses were attributable to dehydration as both salts had 1.17% and 1% hydrates in their moiety chemical formulae, respectively. The malonate (C_32_H_32_BrN_2_O_2_, 2(C_3_H_3_O_4_), C_3_H_8_O) salt was not analyzed with TGA because it is an anhydrous salt. Furthermore, the monohydrate bedaquiline hydrochloride salt (C_32_H_32_BrN_2_O_2_, 2(C_3_H_6_O), Cl, H_2_O) was not analyzed because of the potential of the HCl gas released from the salt to damage the TGA equipment at high temperatures.

#### 3.1.4. Water Sorption Potentials and Water Determination

Hygroscopicity in pharmaceuticals is used to describe the moisture intake capacity of materials. This parameter affects the physicochemical properties of salts, as well as their stability. In this study, we employed two methods: a long-term experiment, using 0% RH and 75% RH at 25 °C, and a short-term procedure with increasing RH chambers of 32.8%, 57.6%, 75.3%, and 92.3%. The long-term sorption/desorption data of the samples suggested that the malonate and hydrochloride salts gained >2% moisture. However, the short-term experiments excluded both salts. The long-term sorption results indicated that measures to mitigate the ingress of moisture may be required in formulating the malonate and HCl salts, and may also shorten their shelf lives. Conversely, the benzoate and nicotinate salts were more stable in the presence of moisture, with gains <2%. Knowledge of hygroscopicity is important in selecting the disintegrants and binders, as well as the humidity controls for the storage conditions of the formulation [8]. A hygroscopic salt can potentially absorb moisture from other excipients in a formulation or capsule shells and this can lead to product failures. However, if the final formulation is an injection, hygroscopicity is not a critical determinant for salt selection [6].

Some of the bedaquiline salts had water adsorbed within their crystal lattices. The Karl Fischer experiments further confirmed the hydration states of the benzoate and nicotinate salts. The information on the hydration states of these salts is important for characterizing them and setting the specifications for their identification. Due to sample size limitations, the water content for the hydrochloride salt was not assessed.

### 3.2. Stability of the New Bedaquiline Salts

Accelerated stability studies provide information related to the ability of a compound to retain its physicochemical properties under high temperatures (40 °C) and moisture (75% RH) over prolonged periods of at least 6 months. The physical and chemical properties of the salts after 3 and 6 months were compared to their forms at the initial 0-month time point. In the protocol of this study, we assessed the salts’ physical stability by means of a visual examination, their chemical structural integrity by means of PXRD, their melting points by mans of DSC, their hydration states by means of TGA, and their potency by means of HPLC analysis.

All salts maintained a white to off-white appearance at the end of 6 months. The PXRD for the benzoate and malonate salts did not show any changes. This suggests that the benzoate and malonate salts were chemically structurally stable under accelerated conditions. However, the hydrochloride salt exhibited altered PXRD values in the 3rd month. This was probably attributable to its low crystallinity, which was determined by comparing the relative contributions of its lattice and activity coefficient to the salt’s aqueous solubility. The diffractogram data indicated that the hydrochloride salt lost its chemical structure after 3 months of accelerated stability, and the 6-month PXRD Rietveld refinement further confirmed the 3-month data.

We encountered some difficulties with the single-crystal measurements for the nicotinate salts; therefore, the Rietveld refinement results of the stability samples were not included in the final scorecard. This challenge was due to the appearance of a PXRD result that differed from the initial single crystal for the nicotinate salt, which suggested the existence of a different polymorphic form. The investigation of this phenomenon will form part of the focus of future studies.

We observed a drop in potency for all the new salts, with the smallest loss observed for the benzoate salt (90.4% to 86.1%), and the greatest loss observed for the hydrochloride salt (101.95 to 77.3%). Intermediate losses of potency were recorded for the malonate (95.6% to 71.3%) and nicotinate (84.7% to 64.1%) salts. In this study, we recognized that some experimental errors may have arisen in assessing the potencies using ~5 mg weights of the salts for the HPLC analysis. The ideal approach would have been to use a microbalance for the small weights (~5 mg) used in the HPLC analysis [22]. In addition, different samples of varying batch sizes, <1 kg, were used for these studies, which offered limited test materials for the stability studies. Therefore, there were variabilities in purity between each batch.

The positions of the DSC endotherms for the 0- and 6-month stability samples for all salts were similar. The positions of the TGA events for the benzoate and nicotinate salts were also similar. The positions of the thermographs suggested that the benzoate and nicotinate salts maintained their hydration, whereas the amount of weight lost was related to the stoichiometry elucidated based on the chemical formulae of the salts [6].

### 3.3. Polymorph Screening of Benzoate Salts

It is important to determine the most thermodynamically stable form of salts that exist in different forms. This is usually achieved by performing polymorph screening. The benzoate salt, which was the lead candidate based on the bedaquiline salt screening, was subjected to different solvent systems to investigate the potential existence of other forms. No new polymorphs were discovered at the 1 kg scale used for this study. The full details of the experiments using the benzoate salt were published in a study on the salts and polymorph screening of bedaquiline [15].

### 3.4. Selecting Benzoate as the Leading Candidate in Bedaquiline Salt Screening

The selection parameters used for choosing a leading salt candidate were based on the principles of ICH Q6. The benzoate salt was selected based on its physicochemical properties as determined in the universal and specific tests conducted on the new bedaquiline salts. The acceptance criteria for each test were based on some widely accepted literature-recommended specifications cited in other pieces of similar research in pharmaceutical drug R&D. Specifically, Aventis [23], Pfizer Global [24], and the general chapters of the USP/NF compendial advisory for the test parameters used in this study.

Although Lipinski’s rule favors salts with a molecular mass <500 Daltons [25], among the new salts, the benzoate had the lowest molecular weight. Its high solubility in 0.01 N HCl (0.3175 mg/mL) was more biorelevant than its observed low aqueous solubility value (4 × 10 − 4 mg/mL). Its low span value (3.7) suggested that with optimized formulations, the flowability properties of the powder would not likely segregate as there were no widespread variations in the particle size distribution. The benzoate had the lowest Hoover-capillary-measured melting point (128 °C ± 1 °C), but this value was still within the acceptable range (≥100 °C) recommended by the Aventis R&D pharmaceutical research organization [23] and the benzoate salt occurred as a hydrate with a well-defined stoichiometric amount of water (C_32_H_32_BrN_2_O_2_, C_7_H_5_O_2_, 1.166(H_2_O)). Although some studies highlighted the risks of dehydration with other hydrates [26,27], the TGA results of the 6-month accelerated samples seemed to suggest that the benzoate salt retained a stoichiometric amount of water in its crystalline lattice. The salt was non-hygroscopic after short-term exposure to a high 92.3% RH and long-term exposure to 75% RH. Therefore, bedaquiline benzoate would not require special protection during manufacturing, handling, and storage. Furthermore, after 6 months, the salt maintained its crystalline structure under accelerated stability conditions of 40 °C and 75% RH, with a minimal loss of chemical potency. Finally, the benzoate salt had the easiest route of synthesis with a high yield; it was easily scaled from 100 mg to 10 kg.

Summarily, the rank order in the choice of a lead candidate in the screening of bedaquiline salts was as follows: benzoate > malonate > nicotinate > hydrochloride.

## 4. Materials and Methods

To obtain the free base from the fumarate salt, a dichloromethane (CH_2_Cl_2_) solution was used to extract the base (three times) using a separatory funnel. Each mixture was washed with saturated sodium bicarbonate (NaHCO_3_) solution. The dichloromethane extract was evaporated over 24 h to remove excess solvents; the residue yielded the amorphous material of the bedaquiline free base [3,28].

For the salt screening experiments, based on the pKa of bedaquiline, 15 acids were selected as potential salt formers. Using bedaquiline patent information, we determined the weights of the individual acids that would form salts with 30 mg of the free base. The target was to achieve the formation of salts using a 1:1 ratio of 30 mg (0.054 millimoles) bedaquiline and the millimole equivalents of the corresponding acids. The solvent selection process was conducted based on the solubility of 30 mg bedaquiline in some ICH class 2 and 3 solvents. The free base and acids were dissolved in 5 mL scintillation vials containing various solvents: acetone, acetonitrile, 2-propanol, water (antisolvent), etc. a complete list of solvents used can be found in Table A1 in the Appendix A of this document. The experimental conditions used in the salt screening included slow and fast evaporation, heat dissolution (supersaturation) and cooling to precipitate salts, and the use of water and hexane as antisolvents. The experiments were placed in a fume cupboard and observed for crystal growths. The single crystals extracted from the mixture of solids were coated with a thin film of Fomblin oil and transferred to the goniometer head of a Bruker Quest diffractometer with a fixed chi angle, a Mo Kα wavelength (λ = 0.71073 Å) sealed-tube fine-focus X-ray tube. The instrument was fitted with a single crystal curved graphite incident-beam monochromator and a Photon100 or Photon II area detector. The full details of the experimental procedure are outlined in our communication on the salts and polymorph screening of bedaquiline [15]. The powder X-ray diffractograms of the salts were obtained on a Panalytical Empyrean Powder X-ray Diffractometer fitted with Bragg-Brentano HD optics, fitted with a sealed-tube copper X-ray source (λ = 1.54178 Å), and a PixCel3D Medipix detector. The sample intensities were measured over a 2θ angular domain from 4° to 40° θ under ambient conditions using the Panalytical Data Collector software. The full details of all experimental procedures are detailed in the salts and polymorphs paper [15].

The principles of ICH Q6 were used to characterize the new salts. High-performance liquid chromatography (HPLC) was carried out to determine the assay results and solubility of the salts. For the assay sample preparation, approximately 5 mg of all samples were dissolved in 10 mL methanol and 1 mL was diluted to 10 mL with the same solvent. The standard preparation was bedaquiline free base (BQ) dissolved and diluted in methanol to obtain a 5.1 µg/mL solution. Regarding the instrumentation of the chromatographic system: an Agilent Technologies 1260 Infinity II, 1200 Series HPLC system was used. Injection volume = 10 microliters, flowrate = 1.2 mL/minute, column temp = 40 °C, Mobile phase: 10 mM ammonium acetate buffer (779 mg Ammonium acetate + 1000 mL, adjust to pH 4.5 with glacial acetic acid) (A), methanol (B), ratio A:B (15:85). A C18 column with with dimensions 250 × 4.5 mm and 5µm was used. PDA detector’s spectral range was 210–400, and chromatographs were extracted at 226 nm [11].

The concentrations of the salts in various solvents were determined using the values of the concentrations and peak areas of the reference standard (bedaquiline free base in methanol) and the peak areas of the salt samples in the solubility medium, as expressed in Equation (1).
(1)$$\frac{\mathrm{Peak}\;\left(\mathrm{salt}\right)}{\mathrm{Peak}\;\left(\mathrm{standard}\right)}\;\times \;\frac{\mathrm{Conc}\;\left(\mathrm{standard}\right)}{\mathrm{Conc}\;\left(\mathrm{salt}\right)}\;\times \;100$$

For particle size and shape determinations through optical microscopy, the Malvern Morphologi G3-ID system utilized automated static imaging features to measure the size distributions of particles. High-magnification microscopy (20× magnification optics) was employed as a technique to analyze particle images. The particle size parameters measured were circle equivalent (CE) diameters. The uniformities in the particle size distribution for the bedaquiline salts were determined based on the particle size values D_10_, D_50_, and D_90_ and were calculated using the span (D_90_ − D_10_)/D_50_. A 5 mm spatula was used to place the salts in the sample holder, and a Nikon A1 Confocal optical microscope with 20× magnification was used to elucidate the particle shapes.

Structural identification with Raman spectroscopy was performed using the Morphologi G3-ID, which offers a unique capability for the chemical identification of individual particles, using morphologically-directed Raman spectroscopy to obtain the Raman spectra for the various bedaquiline salts.

Thermal analyses of melting characteristics and thermal stability were undertaken using differential scanning calorimetry (DSC), thermal gravitational analysis (TGA), and a traditional melting point apparatus. For DSC, a Q10-0072-DSC Q10 series system was used to observe the phenomena associated with the melting of the samples. The samples were placed in holed, sealed aluminum pans coupled to empty reference aluminum pans, which were holed and similarly sealed. The samples were heated from 25 °C to 250 °C at a scanning rate of 10 °C/min.

The changes in the masses of the particulate samples as a function of temperature were determined using a PerkinElmer Thermogravimetric Analyzer (TGA) heated from 25 °C to 250 °C at a rate of 10 °C/min. The instrument temperature and enthalpy adjustments were performed using indium, tin, zinc, and phenyl salicylate, and were finally verified with indium.

A Thomas Hoover capillary melting point apparatus was used to determine the melting ranges of the crystalline salts and the uncorrected values were presented.

The water contents for the hydrated salts were determined using a Metrohm 831 Karl Fischer coulometer attached to a 703 Ti stand titration vessel. The solvent system was Hydranal Coulomat AG, manufactured by Honeywell Fluka. The weights of titrated samples were between 30 mg to 50 mg.

To assess the salts’ water sorption potentials, in a long-term study, accurately weighed salt samples were stored at an ambient temperature of 23 °C under 75% and 0% relative humidity (RH). The conditions for storage were obtained using a saturated brine solution (NaCl) and Drierite for the 75% and 0% RH conditions, respectively. The samples were weighed at regular intervals and the gains or losses in sample weights were recorded routinely. The percentage of weight gain or loss was determined by comparing the differences in the weights of salts compared to the initial sample weight prior to exposure to the abovementioned conditions [9,14]. Short-term assessments were conducted on the new salts using the reference fumarate as a comparator. Saturated solutions of magnesium chloride (32.78% RH), sodium bromide (57.57% RH), sodium chloride (75.29% RH), and potassium sulfate (97.30% RH) were used as chambers to assess the change in weight/the weight of the new bedaquiline salt samples [22].

For the benzoate salt, polymorph screening was conducted using solvents of different polarities. Evaporation, precipitation, crystallization, slurring, and melt crystallization attempts were the experimental conditions studied in the polymorph screening process. Full details of the polymorph screening for the benzoate salt have been documented in a previous publication [15].

In the accelerated stability testing, replicate bedaquiline salt samples were exposed to accelerated conditions in a 75% RH chamber placed inside a Thermo Scientific Heratherm incubator maintained at 40 °C, with an external verifiable temperature readout device. The samples were withdrawn and assessed at 0, 3, and 6-month intervals. Powder X-ray diffraction (PXRD) measurements of the samples were conducted to determine if the salts retained their structural form under accelerated stability conditions. Rietveld refinements were performed against the models of the single-crystal structure datasets using the HighScore software from the Panalytical PXRD instrument. The potencies of the salts were assessed using HPLC chromatographic methods and conditions in the solubility experiments [10]. However, for determinations of potency, the salt samples were dissolved in methanol to achieve a concentration of ~50 µg/mL. Concomitantly, a similar concentration of the bedaquiline free base was used as a reference for the potency determinations. The results were calculated using Equation (1).

## 5. Conclusions

The single crystals of four out of the five salt candidates, namely, benzoate, malonate, nicotinate, and hydrochloride, were successfully elucidated and characterized based on bedaquiline salt screening. The salts were formed using a 1:1 ratio of 30 mg (0.054 millimoles) of bedaquiline and the millimole equivalents of the corresponding acids. The salts were characterized using guidance from ICH Q6 recommendations for creating specifications for new pharmaceutical substances. The results from the general and specific tests enabled us to create preliminary specifications to control the quality of the new bedaquiline salts synthesized in this study. The benzoate (3.7) and hydrochloride (2.5) salts had the lowest span values, suggesting a lower tendency to segregate during powder flow in a compression hopper. The DSC melt values for the malonate (155 °C), HCl (173 °C), benzoate (116 °C), and nicotinate (131 °C) salts were within the acceptable melting temperature ranges for pharmaceutical APIs. Although there was a drop in potency for all new salts, the benzoate salt (90.4% to 86.1%) was the most stable after 6 months under accelerated stability conditions—40 °C/75% RH—after 6 months. Furthermore, the 1.17 hydrated form of the benzoate salt (~3.33% water) and the nicotinate salt (~3.53% water) were both stable to dehydration.

The salts were ranked in the following order: benzoate > malonate > nicotinate > hydrochloride, based on the physicochemical properties of the molecule with the highest developability status. The leading candidate salt, benzoate, was presented as a stable hydrate with no new polymorphs at the laboratory scale of 10 kg. The salt has the potential to be marketed as an NME to obtain FDA approval for treating multi-drug-resistant TB in adult patients.

## Figures and Tables

**Figure 1 pharmaceuticals-16-00257-f001:**
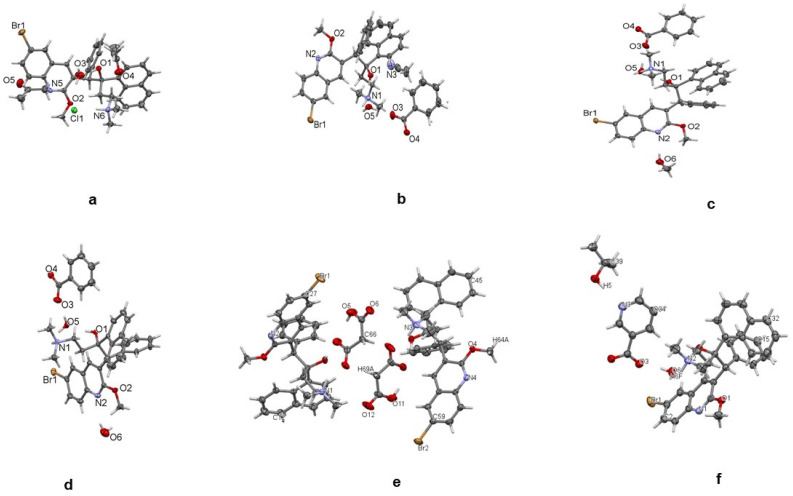
Single crystals of bedaquiline salts. (**a**) HCl, (**b**) benzoate acetonitrile solvate, (**c**) benzoate methanol solvate, (**d**) benzoate hydrate, (**e**) malonate, (**f**) nicotinate.

**Figure 2 pharmaceuticals-16-00257-f002:**
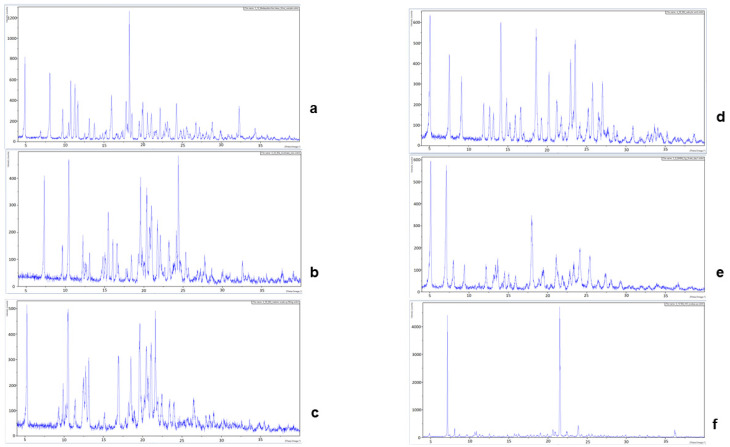
The samples’ intensities, measured over a 2θ domain from 4° to 40° θ, generated distinctive PXRD patterns for the bedaquiline free base and its salts. (**a**) Bedaquiline free base, (**b**) nicotinate, (**c**) malonate, (**d**) salicylate, (**e**) benzoate hydrate, (**f**) HCl.

**Figure 3 pharmaceuticals-16-00257-f003:**
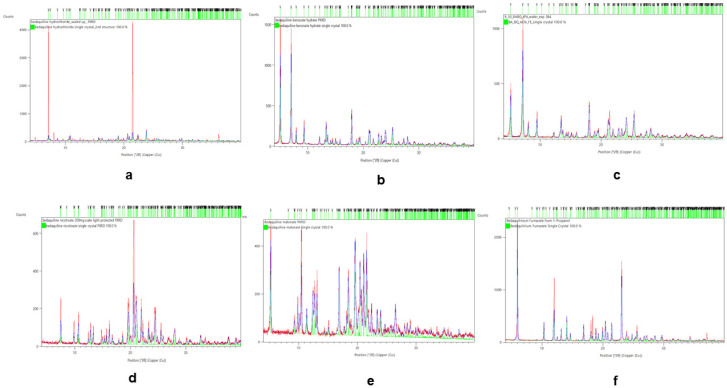
Rietveld refinements for bedaquiline salts confirmed that their PXRD values matched those of their corresponding individual single crystals. (**a**) HCl, (**b**) benzoate hydrate from acetone, (**c**) benzoate hydrate from 2-propanol-water antisolvent, (**d**) nicotinate, (**e**) malonate, (**f**) fumarate.

**Figure 4 pharmaceuticals-16-00257-f004:**
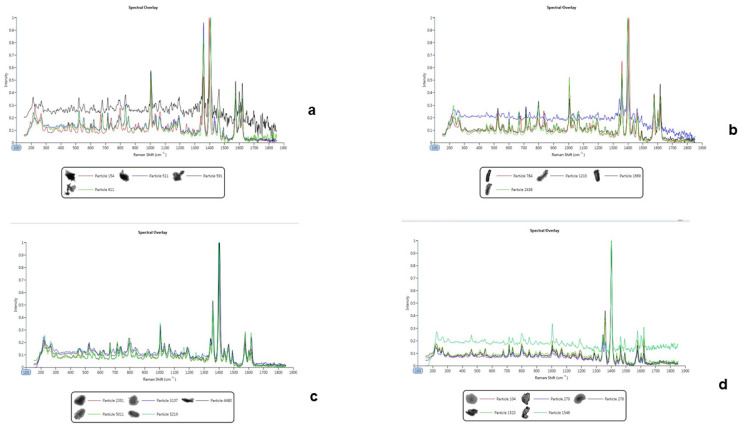
Raman spectra of the free base (uppermost spectra in each overlay) superimposed on the spectra generated by particles of the bedaquiline salts. Therefore, the Raman spectra from the salts could not be differentiated from the free base fingerprint. (**a**) Benzoate salt, (**b**) free base, (**c**) HCl salt, (**d**) malonate salt.

**Figure 5 pharmaceuticals-16-00257-f005:**
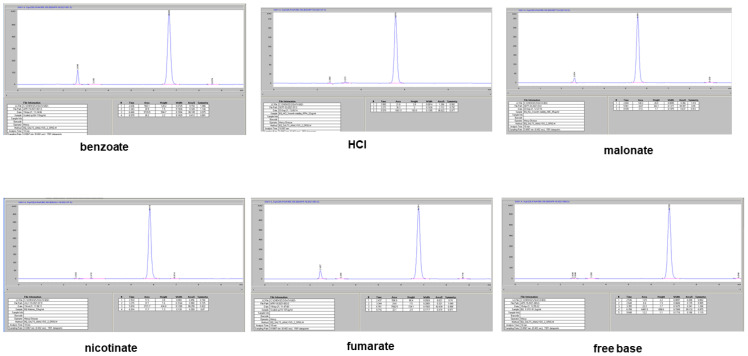
The HPLC retention time sfor the bedaquiline free base were similar to those of the commercially available fumarate salt and the new molecules (5.5 ± 1.2 min).

**Figure 6 pharmaceuticals-16-00257-f006:**
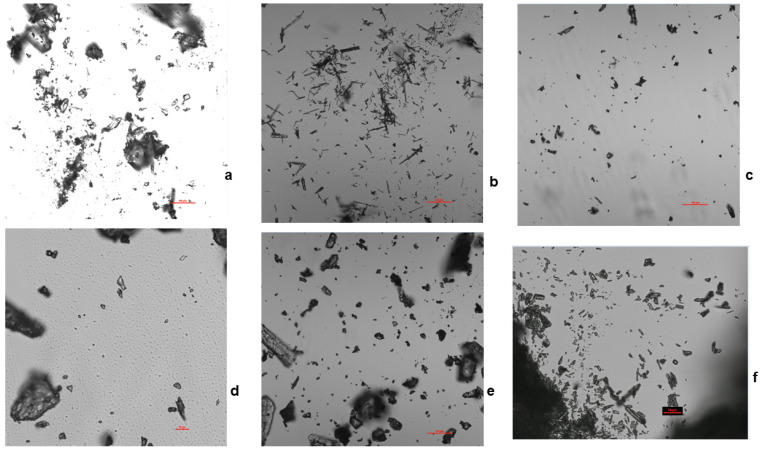
Confocal microscope morphology of particle shapes for bedaquiline salts. At 20× magnification, the images for the HCL and salicylate salts were obtained at a scale of 10 µm; the free base and other salts were measured with a scale of 100 µm. (**a**) The free base consisted mostly of plates, (**b**) benzoate took the form of blades and needles, (**c**) nicotinate was observed as plates and chunks, (**d**) HCl was blades and plates, (**e**) malonate was blades and plates, and (**f**) salicylate was plates and blades.

**Figure 7 pharmaceuticals-16-00257-f007:**
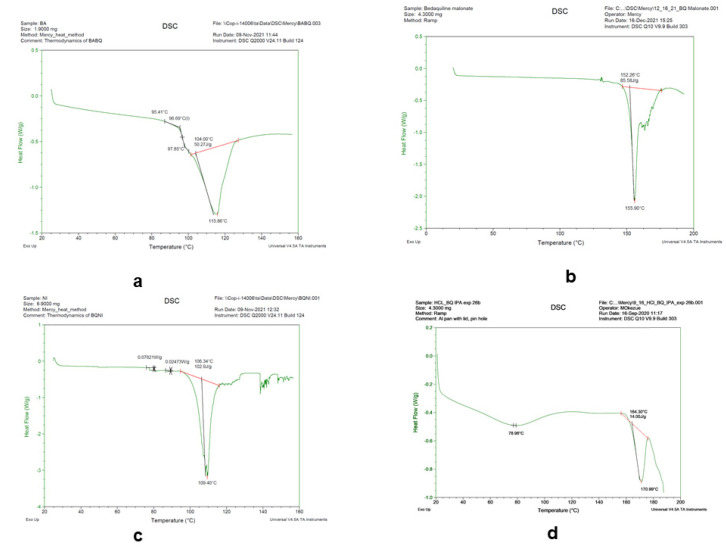
DSC endotherms for bedaquiline salts. (**a**) Endothermic maxima at 98 °C and 104 °C, corresponding to solvent evaporation and melting of benzoate salt prepared from acetonitrile; (**b**) malonate, showing melting point endotherm at ~152 °C; (**c**) nicotinate, showing hydrate loss at 102 °C, melting point endotherm at ~130 °C; (**d**) HCl, showing vapor loss at an initial dip, and melting point endotherm at ~164 °C.

**Figure 8 pharmaceuticals-16-00257-f008:**
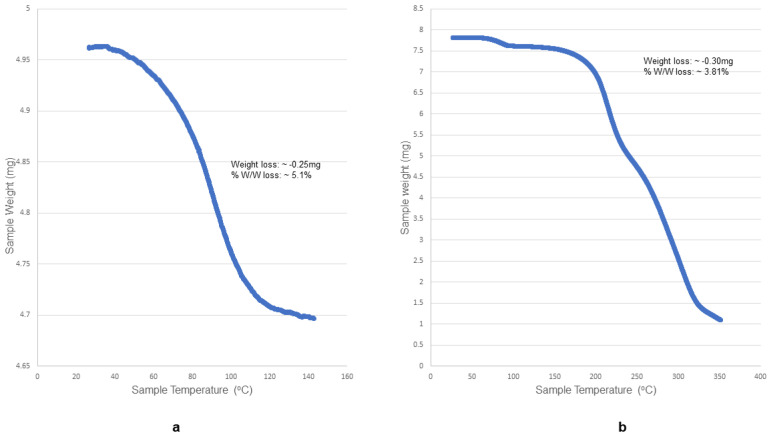
TGA curves showing the weight loss and % *w*/*w* loss for the hydrate nicontinate and benzoate salts. (**a**) TGA for BQ nicotinate, lost ~0.25 mg of its weight when the sample was heated up to 140 °C. (**b**) TGA for BQ benzoate, lost ~0.30 mg of its weight when the sample was heated up to 140 °C.

**Figure 9 pharmaceuticals-16-00257-f009:**
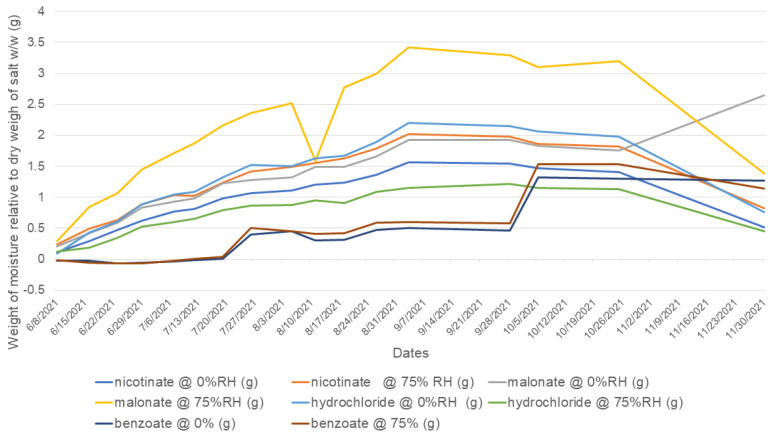
Vapor sorption and desorption trends observed for bedaquiline salts exposed to 0% RH and 75% RH. After 150 days of exposure, malonate and hydrochloride salts gained or lost more moisture than the other bedaquiline salts investigated.

**Figure 10 pharmaceuticals-16-00257-f010:**
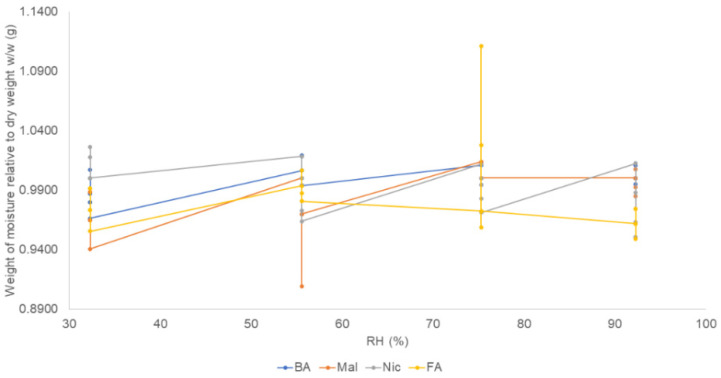
Short-term vapor sorption/desorption trends for bedaquiline salts exposed to increasing RH conditions of 32.8%, 57.6%, 75.3%, and 92.3%. After 21 days of exposure, the weights of moisture relative to dry weight were <2% *w*/*w* (g), suggesting that all salts’ weight changes remained below 0.02% *w*/*w*.

**Figure 11 pharmaceuticals-16-00257-f011:**
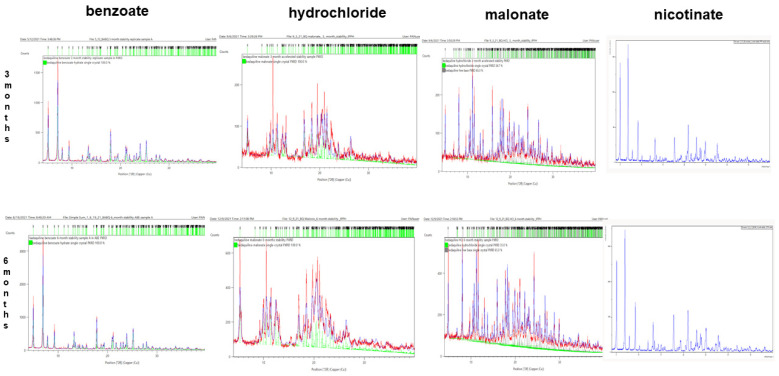
Rietveld refinement of PXRD for 3 and 6- months accelerated stability samples of bedaquiline salts. Over 3 months, benzoate, malonate, and nicotinate salt samples maintained their crystal structures but the hydrochloride salt sample lost its crystallinity. Over 6 months, benzoate, malonate, and nicotinate salt samples retained their crystalline structures but the hydrochloride salt sample lost its crystallinity.

**Figure 12 pharmaceuticals-16-00257-f012:**
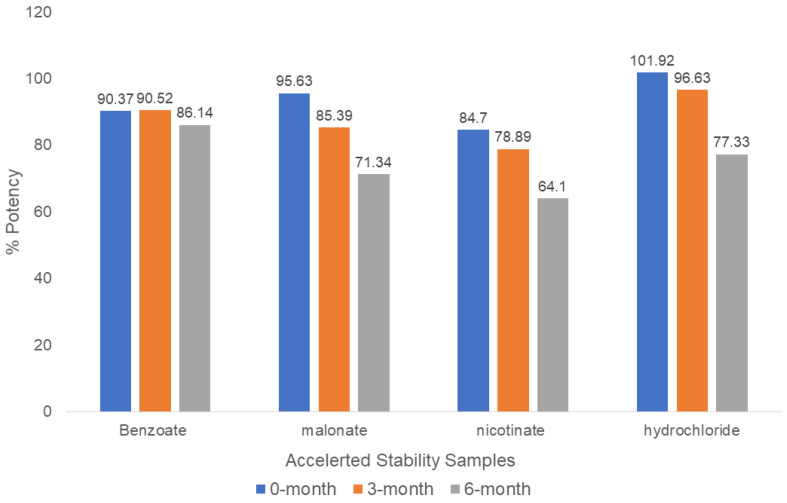
Potencies determined in the accelerated stability studies for bedaquiline salts. The potencies of all bedaquiline salts declined under accelerated studies.

**Figure 13 pharmaceuticals-16-00257-f013:**
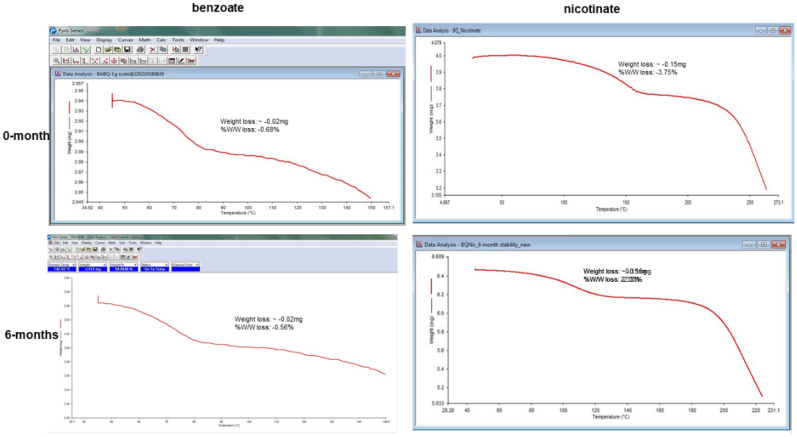
TGA events of dehydration for the initial and 6-month stability samples of benzoate and nicotinate, suggesting that the salts were still hydrated after the stability studies.

**Figure 14 pharmaceuticals-16-00257-f014:**
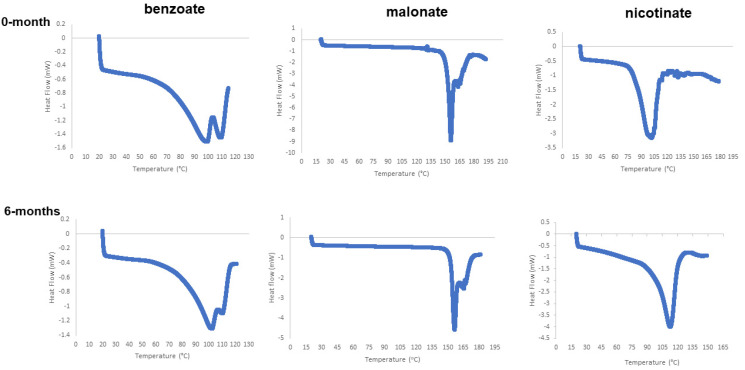
DSC results for 6-month stability samples of new bedaquiline salts closely matched the endotherms observed at the start point (0 months).

**Table 1 pharmaceuticals-16-00257-t001:** Acids (salt formers) and equivalent weights used in the synthesis process.

Acid (Salt Former) Used	Acid (Weight, mg) Equivalent to 30 mg Bedaquiline Base (0.054 Mmoles)	pKa	Solvents Used	Experimental Conditions
Benzoic acid	6.6	4.2	Acetone, 2-propanol, water as antisolvent	Slow evaporation and antisolvent
HCl aq.	1.97	−6.3	Acetone	Slow evaporation
Malonic acid	5.62	2.85	Acetone, and 2-propanol	Slow evaporation
Nicotinic acid	6.65	2.79	Acetone, and 2-propanol	Slow evaporation
Salicylic acid	7.46	2.79	Acetone, and 2-propanol	Slow evaporation

**Table 2 pharmaceuticals-16-00257-t002:** Purity determination results for bedaquiline salts using HPLC analysis.

Salt	Conc Used (µg/mL)	Peak Area	Salt Mwt	Mwt. of BQ	Conc of BQ in Salt Used (µg/mL)	(P_s_/P_std_) * (C_std_/C_s_) * 100
benzoate	51	3089.9	698.7	555.504	40.55	98.3
malonate	53	3721.7	1379.21	555.504	50.68	94.7
HCl	106	8295	726.13	555.504	81.09	132 *
nicotinate	130	8085.4	756.72	555.504	95.43	109.3
BQ base (std)	107	8294.7		555.504		

* Rietveld’s refinement of HCl salt suggested there was a mixture of ~70% salt and ~30% free base. Possible error in purity determination.

**Table 3 pharmaceuticals-16-00257-t003:** Circle equivalent (CE) Diameter (Diam)-number distribution data for bedaquiline salts, using 20× magnification optics.

Salt	CE Diam. Min. (µm)	CE Diam. Max. (µm)	CE Diam. Mean (µm)	CE Diam. STDEV (µm)	CE Diam. (µm) D_10_	CE Diam. (µm) D_50_	CE Diam. (µm) D_90_	Span (D_90_ − D_10_)/D_50_
free base	0.54	130.14	4.29	10.21	0.54	0.67	17.6	25.5
benzoate	0.54	115.1	9.83	9.83	0.64	6.24	23.87	**3.7**
HCl	0.54	75.34	I.87	3.13	0.6	1.07	3.26	**2.5**
malonate	0.54	129.96	9.02	12.14	0.56	1.7	22.11	12.7
nicotinate	0.54	102.46	4.98	10.91	0.54	0.7	15.03	20.7
salicylate	0.54	66.29	5.95	9.51	0.56	0.85	16.84	19.2

**Table 4 pharmaceuticals-16-00257-t004:** Melting ranges obtained for different salts from study experiments.

Salt Experiment	Melting Range (°C)
Bedaquiline benzoate (BABQ) from acetone	128 ± 1
Bedaquiline base	174 ± 1
Benzoic acid	121 ± 1
Bedaquiline hydrochloride salt from IPA slow evaporation (SE) experiments	163 ± 1
Bedaquiline nicotinate from acetone/acetonitrile (ACN) SE	131 ± 1
Bedaquiline malonate from acetone/ACN SE	154 ± 1
Bedaquiline salicylate from acetone/ACN SE	143 ± 1

## Data Availability

Data is contained within the article.

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
