# Peer review of "Synthesis, Characterization, and Stability Assessment for the Benzoate, Hydrochloride, Malonate, and Nicotinate Salts of Bedaquiline"

_pharmaceuticals, 2023, doi:10.3390/ph16020257_

Round 1

Reviewer 1 Report

I just read the article titled Synthesis, Characterization, and Stability Assessment for the Benzoate, Hydrochloride, Malonate, and Nicotinate salts of Bedaquiline submitted to the Journal Pharmaceuticals. The authors investigated the synthesis of Bedaquiline, a type of active pharmaceutical product, in the presence of benzoate, nicotinate, and malonate salts. The obtained products were characterized detailed using XRD, Raman, Malvern Morphologi G3, microscopy, and TGA/DSC to determine the structure, morphology, and thermal characteristic of the crystals.

-The manuscript generally presents new results and is well-written; however, the main issue is related to the quality of the Figures. Before publication, it needs to be improved (especially Figs 2, 3, 4, 5, 7, 8, 11, 13). I tried to understand/examine the Figures but was not possible as some figures needed to be more apparent.  In addition, I could not see the text/data on the Figures.

- XRD results:  Please first add the XRD result of the Bedaquiline (not including any salts). At least we can compare the results obtained in acid media.

-Please add a scale bar in Figure 6 for each microscopic image. The images do not reflect the shape of the crystals. In addition, the microscopic images and Morphologi G3 (Table 4) results do not support each other. Please check! I suggest adding SEM images instead of a Confocal microscope. At least the authors can use the images obtained from the Morphologi G3 device.

-Please also add DTG curves for hydrate salts.

-Please also add some quantitative results in the Conclusion part.

Reviewer 2 Report

The present study concerns the synthesis and investigation of a number of bedaquiline salts, a promising compound for the treatment of tuberculosis. Of course, the work makes a great contribution to the field under study and is very valuable. As a small remark, I recommend careful handling of the term "weak acids" in the abstract, since one of the reagents is hydrochloric acid.

Author Response

Response 1:

Thank you so much for the observation, “counter ions (acids)” now replaces the word ‘weak’ in the Abstract section 

Reviewer 3 Report

Dr. Okezue and Prof. Byrn have extended their studies to elucidate the structure of bedaquiline and are reporting the synthesis, characterization and stability of four additional new potential molecular entities, namely benzoate, hydrochloride (HCl), nicotinate, and malonate salts of bedaquiline. The studies are interesting enough to be published in Pharmaceuticals. However, I recommend the authors to address the following points in their revised manuscript.

1. X-ray characterization of the reported salts is an imported part of the manuscript but no detailed discussion of their bond angles and lengths have been added. Surprisingly no checkcifs files have been provided or R-values have been mentioned in the manuscript.

2. CCDC number of the reported compounds must be included in the revised version.

3. NMR is a common characterization technique, I would recommend to add the 1H and 13C NMR data of the reported compounds.

4. Figures of crystals and their labeling in Figure 1 are too small to be seen. I would recommend improving it in the revised version.

5. Since, the crystal structures are in hand, it might be interesting to apply docking studies to evaluate their interaction with host enzymes.

6.  Refinement details such as R-vlaues, measurement temperature should be added to Table 2.

7.  Authors should add a ChemDraw figure of the reported compounds and use numbers instead of using their full names repeatedly in the text.

8.  Manuscript is too long; I would recommend to move for instance Table 2 and B1 to supporting information.

9.  Abbreviations such as KF should be explained.

10. Line 379, “Forming salts of a poorly aqueous molecule may improve its oral bioavailability”. I am not sure what the authors trying to say here. If it is a common phenomenon, the authors should add a reference to justify their claim.

11. There are too many formatting errors in the paper that can be avoided. Some of these mistakes are highlighted below. Such as;

i)  Remove word Title from title and keyword1 from keywords.

ii) Please use uniformly mL instead of ml.

iii) Line 65 should be, “Accurate measurements of these parameters are characteristics of pharmaceutical compounds and can be used in their characterization (7).”

iv)  Line 75, “the manufacturing” should be “manufacturing”.

v) Some of the subtitle are beginning with capital letters while others not, so please be consistent.

Round 2

Reviewer 1 Report

It can be accepted in its revised form.

Reviewer 3 Report

I am satisfied with the revision. It can be accepted in the present form, however, the authors should replace the text by reference number [3] in line 126.